# Pb(II) Adsorption Properties of a Three-Dimensional Porous Bacterial Cellulose/Graphene Oxide Composite Hydrogel Subjected to Ultrasonic Treatment

**DOI:** 10.3390/ma17133053

**Published:** 2024-06-21

**Authors:** Xinxing Zhang, Jing Xu, Zhijie Zhang, Pengping Li, Chang He, Mingfeng Zhong

**Affiliations:** 1Key Laboratory of Harbor & Marine Structure Durability Technology, Ministry of Transport of PRC, Guangzhou 510640, China; zxx1206182067@163.com (X.Z.); lpengping@cccc4.com (P.L.); 2School of Materials Science & Engineering, South China University of Technology, Guangzhou 510230, China; g2533616566@163.com (J.X.); imzhang@scut.edu.cn (Z.Z.); 15116645089@163.com (C.H.)

**Keywords:** bacterial cellulose, graphene oxide, ultrasonic method, adsorption, Pb(II)

## Abstract

A three-dimensional porous bacterial cellulose/graphene oxide (BC/GO) composite hydrogel (BC/GO) was synthesized with multi-layer graphene oxide (GO) as the modifier and bacterial cellulose as the skeleton via an ultrasonic shaking process to absorb lead ions effectively. The characteristics of BC/GO were investigated through TEM, SEM, FT-IR, NMR and Zeta potential experiments. Compared to bacterial cellulose, the ultrasonic method and the carboxyl groups stemming from GO helped to enhance the availability of O(3)H of BC, in addition to the looser three-dimensional structure and enriched oxygen-containing groups, leading to a significantly higher adsorption capacity for Pb(II). In this paper, the adsorption behavior of BC/GO is influenced by the GO concentration, adsorption time, and initial concentration. The highest adsorption capacity for Pb(II) on BC/GO found in this study was 224.5 mg/g. The findings implied that the pseudo-second-order model explained the BC/GO adsorption dynamics and that the data of its adsorption isotherm fit the Freundlich model. Because of the looser three-dimensional structure, the complexation of carboxyl groups, and the enhanced availability of O(3)H, bacterial cellulose exhibited a much better adsorption capacity.

## 1. Introduction

Recently, as industries have developed, water system pollution has posed a serious threat to the environment; these industries include metallurgy engineering, chemical engineering, the petrochemical industry, electronic engineering, etc. Lead ions are one of the most representative pollution sources [1]. Once lead ions enter the human body, regular exposure may lead to excessive lead in the blood via blood circulation in the body. Excess lead ions cause damage to various organ tissues, such as the brain, liver and kidneys; the most serious effects occur in the blood system and the digestive system [2].

Therefore, the task of treating lead ions in water is critical and urgent. There are various methods to deal with lead ions. At present, the main treatment methods consist of adsorption, ion exchange, solvent extraction, chemical precipitation, etc. [3,4,5]. Current research reports that adsorption is the most promising, with a low cost, high effectiveness and simplicity [6]. A range of absorbents have been fabricated for the removal of Pb(II), including carbon adsorbents [7], mineral adsorbents [8], polymeric adsorbents [9], biosorbents [10], nanomaterial adsorbents [11] and so on.

Bacterial cellulose (BC), a biosynthesized cellulose, is a prospective biosorbent for Pb(II) elimination owing to its three-dimensional cross-linked network structure, relatively large surface area, and rich hydroxyl groups in its chain [12,13,14]. The adsorption performance of BC mainly depends on the hydroxyl activity of O(2)H, O(3)H and O(6)H in the molecular structure [15]. The accessibility of hydroxyl follows the order O(2)H>O(6)H>O(3)H. In high-crystalline-degree cellulose, O(3)H forms intermolecular hydrogen bonds with O(5) and is poorly accessible; in completely disordered cellulose, the intermolecular hydrogen bonds (O(3)H···O(5)) are partially broken and the accessibility of O(3)H is enhanced [16]. Therefore, not all hydroxyl groups are active, which restricts the adsorption capacity. Huang et al. fabricated a nanostructured amino-functionalized magnetic bacterial cellulose/activated carbon composite bioadsorbent (AMBCAC) for the removal of Pb^2+^. Its uneven surface and amino groups provided more active adsorption sites, exhibiting a high adsorption capacity of 70 mg/g [17]. Liu et al. enriched the oxygen-containing groups of BC by introducing amide and carboxyl groups, enhancing the hydroxyl activity and increasing the adsorption properties of the adsorbent [18]. So, group modifications can effectively improve the adsorption capacity.

In recent years, graphene oxide (GO) has received extensive attention due to its large specific surface area, its planar defects, and the considerable number of oxygen-containing active groups, which provide a wealth of adsorption sites and contribute to lead ion adsorption [19,20,21]. However, the reclamation difficulty and relatively high costs restrict the usage of GO [22]. Therefore, the combination of BC and GO can reduce or avoid the secondary pollution of GO into aqueous solutions. Meanwhile, abundant oxygen-containing groups enrich the adsorption activity of BC [23,24].

In this work, we synthesized a BC/GO composite hydrogel with multi-layer GO as the modifier and bacterial cellulose as the skeleton via an ultrasonic shaking method for Pb(II) removal. Ultrasonication increases the adsorption sites on bacterial cellulose’s surfaces. Rapid magnetic separation of GO was made possible by modifications. The BC/GO composite hydrogel has merits of a high adsorption capacity, environmentally friendliness and a high dispersivity. We employed different test methods to characterize BC/GO and analyze its adsorption capacity. By fitting the kinetics and isotherm adsorption equation, the adsorption mechanism of the materials was explored.

## 2. Materials and Methods

### 2.1. Materials

Bacterial cellulose was supplied by Jinan Jinhuafang Food Technology Co. Ltd., Jinan, China. Graphene oxide was purchased from Suzhou Hengqiu Technology Co., Ltd., Suzhou, China. Pb(NO_3_)_2_, HNO_3_ and NaOH were purchased from Guangzhou Chemical Reagent Factory, Guangzhou, China and used as received in our experiments without further purification.

### 2.2. Preparation of BC/GO Composite Hydrogel

Certain amounts of GO and deionized water were mixed for 90 min via an ultrasonic dispersion method to prepare a solution with concentrations of 2.0 g/L, 0.4 g/L and 0.2 g/L, respectively. BC underwent three rounds of washing in deionized water and was soaked in 4 M NaOH solution in a water bath at 80 °C for 4 h. After soaking, deionized water was used to wash BC until the pH reached 7.0 and the BC was semi-transparent and milky white. Then, BC was compressed, and the BC slurry and GO aqueous solution (2.0 g/L, 0.4 g/L and 0.2 g/L) (volume ratio was 1:1) were mixed into a flask, respectively. After stirring uniformly, the mixture was ultrasonically shaken at 50 °C for 90 min and left to stand for a certain time to obtain the BC/GO composite hydrogel, marked as BC/GO-1 (2.0 g/L), BC/GO-2 (0.4 g/L), BC/GO-3 (0.2 g/L).

### 2.3. Characterization

The information of the adsorbents, determined via FT-IR spectroscopy (ATR), was recorded on a VER TEX70 in the range of 3750–550 cm^−1^. The morphology evolution was observed using a NOVA NANOSEM 430-field emission scanning electron microscope (from FEI USA, Inc., Hillsboro, OR, USA) at an acceleration voltage of 15 kV with a beam of 3.0 and a JEM-2100F transmission electron microscope (from Nippon Electronics Co., Tokyo, Japan). The chemical structure was analyzed via ^13^C-NMR spectroscopy, observed with an AVANCE digital 400. The surface Zeta potential of BC/GO-1 was determined using a Zeta sizer nanoparticle size potentiometer (Nano-ZS90), Malvern, UK.

### 2.4. Adsorption Experiment

Bath adsorption tests were carried out using an agitation speed of 300 rpm and the conventional stirring method at 28 ± 2 °C. All the adsorption experiments were performed in flasks filled with 0.4 g/L adsorbent. To investigate the impacts of the GO concentration on the adsorption of Pb(II), the initial concentration of Pb(II) was 100 mg/L. Adsorption kinetics were determined in solution at pH 5.0, and lead ion concentrations were measured using samples collected at different times. Adsorption isotherm experiments were conducted at pH 5.0 with initial lead ion concentrations in the range of 5–100 mg/L. Additionally, the details are described in the relevant figures. Before the tests, adsorbents were centrifuged from the solution, filtered, and analyzed. An ICP test was employed to ascertain the quantity of lead ions present in the solution.

According to the difference in lead ion concentration, the following equation was used to determine how many metal ions were adsorbed on each unit of adsorbent (1):(1)$$q_{e}=(C_{0}-C_{e})V/W$$
where *q_e_* is the equation adsorption capacity (mg/g), *C*_0_ is the initial concentration (mg/L), *C_e_* is the liquid-phase metal concentration (mg/L), *V* is the volume of metal ion solution (L), and *W* is the weight of adsorbents, calculated by the following Equation (2):(2)$$W=W_{0}\times X$$
where *W*_0_ is the weight of the BC and BC/GO (g) and *X* was obtained by the referring to GB/T 1725-1979 [25].

## 3. Results and Discussion

### 3.1. Characterization of BC/GO

The synthesis process of BC/GO is illustrated in Figure 1. Under ultrasonication, the intramolecular and intermolecular hydrogen bonds in BC will break and GO will peel off to single layers or less. Further heating and stirring can lead to the dehydration reaction of BC with the hydroxy bonds of GO and the condensation reaction of BC with the carboxyl groups of GO.

### 3.2. TEM and SEM Studies

As shown in Figure 2, in the TEM image of the BC/GO-1 composite hydrogel, it is clear that GO is attached to the surface of BC. In order to observe the microscopic structure of the composite hydrogel, BC and BC/GO-1 were freeze dried [26]. A structure with relatively uniform holes guarantees the formation of a looser three-dimensional structure, as shown in Figure 2c. This is because the high-intensity ultrasonic energy made the cavity grow rapidly and it was easier to generate an implosion, which occurred on the cellulose surface and was transmitted along the fiber extension direction [27,28,29]. Meanwhile, GO was peeled off under the action of ultrasound. Then, GO nanosheets and BC fibers formed a looser three-dimensional structure. Therefore, this microstructure provides a large contact surface area for the removal of lead ions.

### 3.3. FT-IR Studies

BC was polymerized to a six-membered ring structure with two asymmetric oxygens linked by a glycosidic bond. The hydroxyls at O(2)H, O(3)H and O(6)H were the main reactive groups in the chemical modification [30]. The split-peak fitting results are shown in Table 1. The peak at 3000–3600 cm^−1^ reflects the intramolecular hydrogen bonds (O(2)H···O(6), O(3)H···O(5)) and the intermolecular hydrogen bonds (O(6)H···O(3)) [31]. The results show that the relative intensities of O(2)H···O(6) and O(3)H···O(5) of BC/GO-1 decreased and those of O(6)H···O(3) increased, suggesting that GO may cross-link with the O2 and O(6) of BC; the partial breakage of O(3)H···O(5) and the enhancement of the activity of O(3)H increased the adsorption of heavy metal ions by BC.

The peaks at 3000–3600 cm^−1^ are split, as displayed in Figure 3b,c. The peak differentiation and imitation results are listed in Table 1; the intensity of BC/GO-1 at O(2)H···O(6) is significantly reduced and the intensity of BC/GO-1 at O(3)H···O(5) and O(6)H···O(3) is enhanced, indicating that GO possibly cross-linked with O(2) and O(6) [32]. Figure 3a shows the FT-IR spectra of GO, BC and BC/GO-1. Compared with GO, a new peak arises in the BC/GO-1 spectra at 1733 cm^−1^, which is assigned to C=O. The C=O bond could result from the carboxyl group of GO or stem from a reaction between the carboxyl group on GO and the hydroxyl group of BC. The peak at 1223 cm^−1^ corresponding to the C-O bond of the phenolic hydroxyl group of GO disappears, implying that a dehydration reaction occurs between the hydroxyl group of BC and the phenolic hydroxyl group of GO, forming a C-O-C group [33].

### 3.4. NMR Studies

Figure 4 demonstrates ^13^C-NMR chemical shifts of BC and BC/GO-1. Compared with the spectrum of BC, two new peaks appear at 167.82 ppm and 172.12 ppm, attributed to a carboxyl group and an ester group, respectively [34]. The carboxyl group stems from the unreacted group on GO, and the ester group is produced from the reaction of GO with BC. The results of this analysis are in accordance with those of the FT-IR pattern analysis, which infers that the species of oxygenic groups of BC/GO-1 are more abundant compared to BC.

The ^13^C-NMR spectrum is sensitive to crystalline and amorphous regions and can be used to analyze the crystallinity of bacterial cellulose. In the ^13^C-NMR spectrum, the chemical shift at the C4 position can be divided into two parts: one is the chemical shift of the crystal region at 86–92 ppm and the other is the chemical shift of the amorphous region at 79–86 ppm. The crystallographic index is calculated by Formula (3) [35]:(3)$$CI^{NMR}=\frac{A_{86-92\;ppm}}{\left(A_{79-86\;ppm}+A_{86-92\;ppm}\right)}\times 100\%$$
where *A* is the peak area corresponding to the chemical shift.

The *CI^NMR^* values of BC and BC/GO-1 were 83.02% and 62.08%, respectively. The mechanical shear power of ultrasonic treatment randomly cleaved the hydrogen bond (O(3)H···O(5)) and increased the availability of O(3)H, which suggests that the crystallinity index of the composite decreased.

### 3.5. Zeta Potential Studies

The electrostatic potential of the diffusion layer in the electric double layer depends on the Zeta potential. As displayed in Figure 5, the surface of BC/GO-1 is electrically negative owing to the ionization of oxygen-containing groups in the pH range of 2.0–8.0, which infers that BC/GO-1 still has an excellent adsorption capacity under acidic or neutral conditions. When the pH is 5.0, the absolute value of the Zeta potential of the BC/GO-1 solution reaches the minimum value; a large amount of dissociation of surface oxygen-containing groups provides effective reactive sites. The following tests were used to determine the ideal pH for Pb(III) adsorption as 5.0.

### 3.6. Adsorption of Pb(II)

#### 3.6.1. Effect of GO Concentration on Adsorption

Figure 6a shows the adsorption capacity of BC, BC/GO-1, BC/GO-2, and BC/GO-3 for a Pb(II) solution. At a pH of 5.0, the adsorption capacity of BC/GO-1 is significantly much higher than that of BC, BC/GO-2, and BC/GO-3. This is indicative of the promotion of the adsorption capacity of BC/GO with the increasing GO content. Thus, BC/GO-1 was selected in the following experiments.

#### 3.6.2. Effect of Contact Time on Adsorption

The adsorption equilibrium time is one of the key metrics for assessing the effectiveness of adsorbents in adsorption. The impact of contact time on Pb(II) adsorption is shown in Figure 6b. The adsorption behavior reaches equilibrium in 90 minutes, and the adsorption amount is 224.5 mg/g, which is much higher than that of BC (121.0 mg/g). From the results of the experiments, the adsorption capacity of BC/GO-1 is significantly improved after the process of modification.

#### 3.6.3. Effect of Initial Concentration on Adsorption

An initial concentration study was conducted to investigate the impact of Pb(II) concentration on the adsorption capacity. The adsorption capacity rises as the Pb(II) concentration rises, as seen in Figure 6c. However, when *C*_0_ is more than 70 mg/L, the rate of increase in the adsorption capacity declines. When there is a difference in Pb(II) concentration between the adsorbent surface and the solution, the lead ions in the solution rapidly move to the surface of the adsorbents, driven by the difference in concentration. The higher the concentration is, the larger the Pb(II) amount is, which enhances the probability of lead ions binding to the adsorption sites on the surface of BC/GO-1.

In addition, compared to maximum adsorption capacity of other modified bacterial cellulose adsorbents for Pb(II) (listed in Table 2), it can be found that BC/GO-1 has a better capacity of 224.5 mg/g, suggesting that BC/GO is a potentially good material for the effective removal of lead ions from aqueous solutions.

### 3.7. Adsorption Kinetics

The kinetic model is an important research approach for studying the adsorption process. In order to study the adsorption behavior of adsorbents of Pb(II), two kinetic models (a pseudo-first-order kinetic model and a pseudo-second-order kinetic model) were used to fit the experimental data [41]. The following equations are used.

The pseudo-first-order adsorption kinetics are as follows:(4)$$\ln(q_{e}-q_{t})=\ln q_{e}-k_{1}t$$
where *k*_1_ is the pseudo-first-order kinetic equation coefficients (min^−1^) and *q_e_* and *q_t_* are the adsorption amount (mg/g) at equilibrium and over time, respectively.

The pseudo-second-order adsorption kinetics are as follows:(5)$$\frac{t}{q_{t}}=\frac{1}{{k_{2}q_{e}}^{2}}+\frac{t}{q_{e}}$$
where *k*_2_ represents the coefficient of the pseudo-second-order dynamic equation (g/(mg·min)). *q_e_*, *k*_1_ and *k*_2_ can be derived from the intercept and the slope of the line.

The fitted linear curves of the two kinetics are shown in Figure 7. The constants of the two kinetic models and the correlation coefficients are shown in Table 3. According to the value of the correlation coefficients, the experimental data of BC and BC/GO-1 were more consistent with the pseudo-second-order kinetic model (*R*^2^ = 0.9757, 0.9817), which indicates that the adsorption rate is controlled by chemical adsorption. Consequently, with an increase in active oxygen-containing group sites, the adsorption capacity improves.

### 3.8. Adsorption Isotherm

The adsorption mechanism of Pb(II) was analyzed to further study the adsorption properties. The Langmuir adsorption isotherm represents single-molecule adsorption, as shown in Equation (6) [42]. The Freundlich adsorption isotherm is an empirical formula and is generally considered to be a multi-layer adsorption [43]. The expression is shown in Equation (7):(6)$$\frac{C_{e}}{q_{e}}=\frac{1}{k_{L}q_{m}}+\frac{C_{e}}{q_{m}}$$
where *C_e_* is the initial concentration (mg/L), *q_m_* and *q_e_* are the equilibrium adsorption capacity (mg/g) and the maximum adsorption capacity (mg/g), respectively, and *k_L_* represents the coefficient of Langmuir adsorption.
(7)$$\ln q_{e}=\ln k_{F}+\frac{1}{n}\ln C_{e}$$
where *k_F_* represents the coefficient of Langmuir adsorption and *n* is an empirical constant of the Freundlich isotherm adsorption model. *q_e_*, *k_L_* and *k_F_* can be derived from the intercept and the slope of the line.

The isotherm linear fitting results are shown in Figure 8. The Langmuir and Freundlich adsorption constants evaluated from the isotherms with the correlation coefficients are listed in Table 4. The experimental data of BC and BC/GO-1 are more consistent with the Freundlich model (*R*^2^ = 0.9983, 0.9944), suggesting that the adsorption behavior conformed well to multi-layer adsorption.

### 3.9. FT-IR Studies

A number of changes in the FT-IR spectrum of Pb(II)-loaded BC/GO-1 (Figure 9) are observed. After Pb(II) loading, the peaks at 3371 cm^−1^ and 1631 cm^−1^ clearly shift to 3343 cm^−1^ and 1624 cm^−1^ (after adsorption—spectrum of BC/GO-1), respectively, which implies that Pb(II) may interact with some hydroxyl groups. The peak at 1733 cm^−1^ corresponding to C=O is much smaller than that without adsorption, implying that this group interacts with lead ions [44]. In addition, the peak at 1575 cm^−1^ is attributed to metal complexation [45].

The adsorption mechanism of Pb(II) is summarized as follows (Figure 10): (1) electrostatic attraction between Pb^2+^ and –OH of the adsorbent and (2) chelation between the C=O group and Pb(II) ions.

## 4. Conclusions

In summary, BC/GO for Pb(II) removal was fabricated via an ultrasonic vibration process with multi-layer GO as a modifier. Via an ultrasonic shaking process, the hydrogen bond (O(3)H···O(5)) was broken, increasing the availability of O(3)H, and GO adhered to the surface of the BC microfibers, forming a looser three-dimensional BC/GO composite hydrogel with a relatively uniform hole, which increased the contact area with lead ions. Meanwhile, the carboxyl group introduced by GO enriched the oxygen-containing groups of BC, which increased the attractive force for Pb(II) removal. At a pH of 5.0, the range of deprotonation of –OH and –COOH was maximized, which provided more effective adsorption sites. According to adsorption kinetics fitting, the primary adsorption of BC/GO was chemical adsorption. The adsorption isotherm fitting results indicated that the adsorption structure is mainly based on multi-layer structure adsorption, and the equilibrium adsorption capacity can reach 224.5 mg/g. This study demonstrates that BC/GO may be a potential alternative for heavy metal ion removal.

## Figures and Tables

**Figure 1 materials-17-03053-f001:**
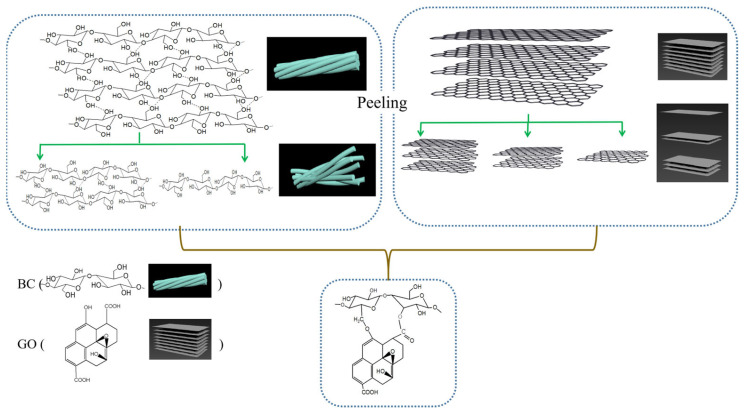
The synthesis process of BC/GO.

**Figure 2 materials-17-03053-f002:**
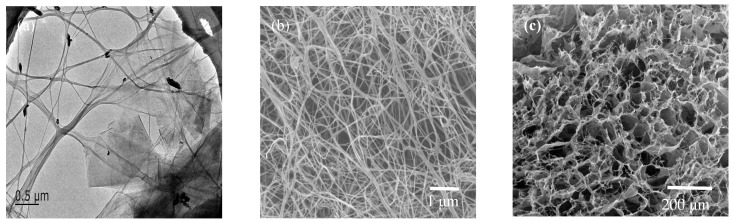
TEM image of BC/GO-1 composite hydrogel (**a**) and SEM images of BC (**b**) and BC/GO-1 (**c**).

**Figure 3 materials-17-03053-f003:**
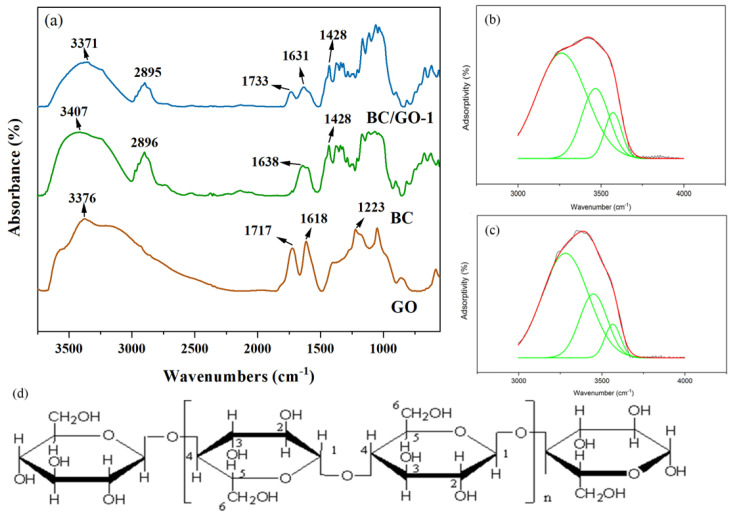
(**a**) FT-IR spectra of GO, BC and BC/GO-1, (**b**) results of peak fitting of BC at 3000–3600 cm^−1^, (**c**) results of peak fitting of BC/GO-1 at 3000–3600 cm^−1^, (**d**) the structure of BC.

**Figure 4 materials-17-03053-f004:**
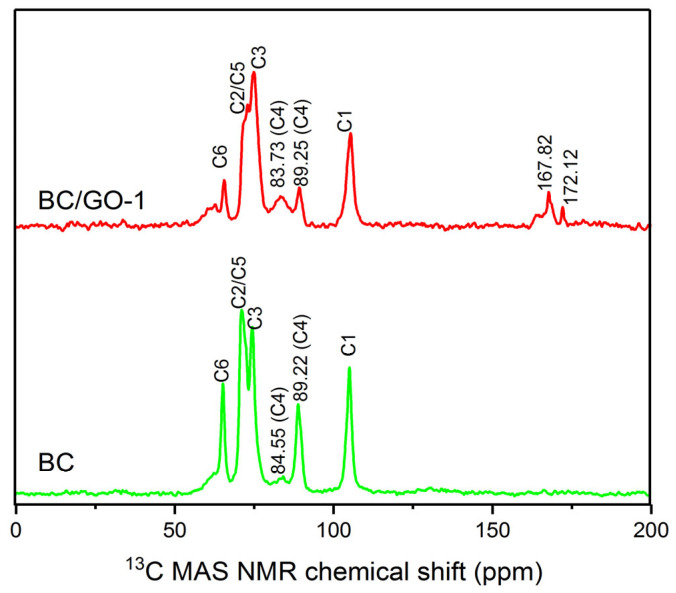
^13^C MAS NMR chemical shift of BC and BC/GO-1.

**Figure 5 materials-17-03053-f005:**
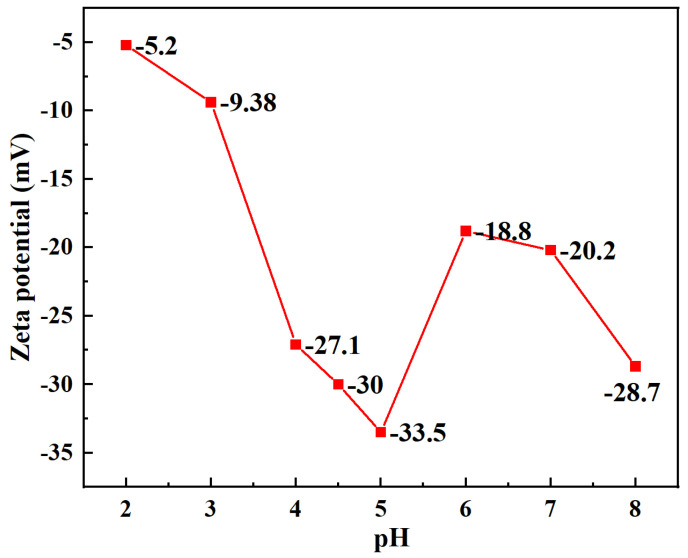
Zeta potential of BC/GO-1.

**Figure 6 materials-17-03053-f006:**
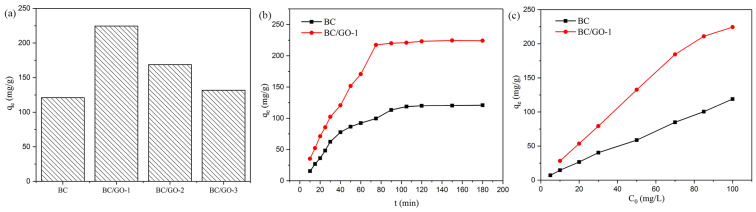
(**a**) The adsorption capacity of BC, BC/GO-1, BC/GO-2, and BC/GO-3, (**b**) effect of contact time on the adsorption of Pb(II) onto BC and BC/GO-1 at an initial pH of 5.0, and (**c**) effect of the initial concentration of Pb(II) on the adsorption capacity onto BC and BC/GO-1; experimental conditions: initial concentration, 5–100 mg/L; sample dose, 0.4 g/L; pH, 5.0; temperature, 28 ± 2 °C; contract time, 90 min.

**Figure 7 materials-17-03053-f007:**
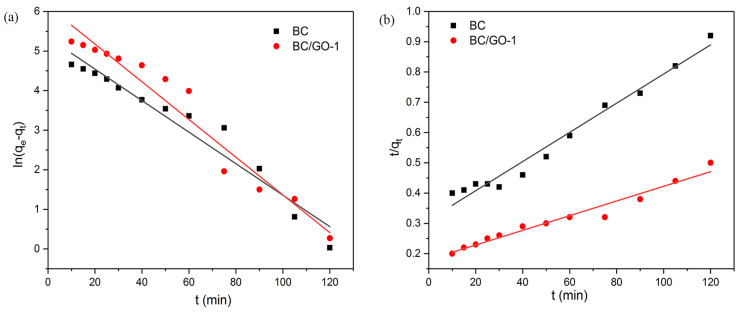
(**a**) Pseudo-first-order and (**b**) pseudo-second-order kinetic models of Pb(II) (sample dose of 0.4 g/L, pH 5.0, temperature of 28 ± 2 °C, C_0_ = 100 mg/L, contract time of 10−120 min).

**Figure 8 materials-17-03053-f008:**
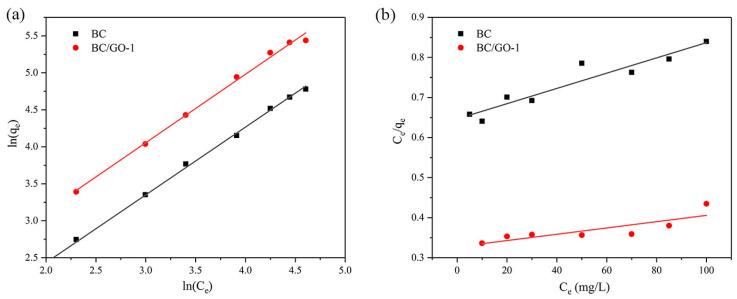
(**a**) Langmuir and (**b**) Freundlich adsorption isotherm fit for Pb(II) (C_0_ = 5–100 mg/L, sample dose of 0.4 g/L, pH 5.0, temperature of 28 ± 2 °C, contract time pf 90 min).

**Figure 9 materials-17-03053-f009:**
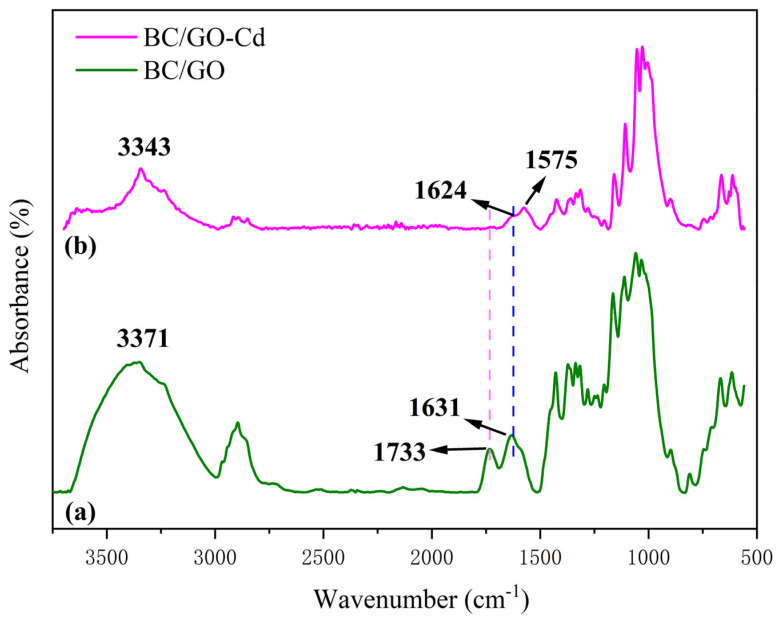
FT-IR spectra of (**a**) BC/GO-1 and (**b**) Pb(II)-loaded BC/GO-1.

**Figure 10 materials-17-03053-f010:**
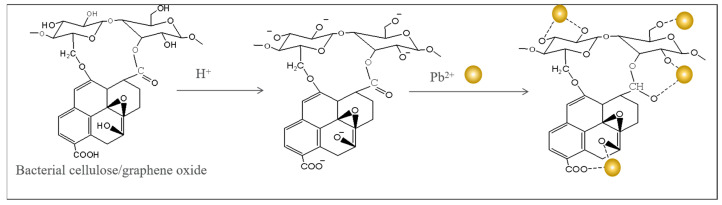
Proposed adsorption mechanism of Pb (II) onto BC/GO-1.

**Table 1 materials-17-03053-t001:** Fitting results of the hydrogen region in FT-IR.

Sample	Wavenumber/cm^−1^ (Area IntgP %)		
	O(2)H···O(6)	O(3)H···O(5)	O(6)H···O(3)
BC	3566 (13.7)	3436 (46.42)	3228 (39.88)
BC/GO-1	3563 (9.55)	3432 (44.01)	3251 (46.44)

**Table 2 materials-17-03053-t002:** Comparison of maximum adsorption capacities of modified bacterial cellulose absorbents for Pb(II).

Adsorbent	Adsorption Capacity (mg/g)	Ref.
Carboxymethylated bacterial cellulose	60.4	[36]
Fe_3_O_4_/BC spheres	65.0	[37]
Amidoximated bacterial cellulose	67.0	[38]
Diethylenetriamine-bacterial cellulose	87.4	[39]
Polyethyleneimine-bacterial cellulose	148.0	[40]
Amino-functionalized magnetic bacterial cellulose/activated carbon	161.8	[17]
BC/GO-1	224.5	This work

**Table 3 materials-17-03053-t003:** Adsorption kinetic parameters for Pb(II) adsorption onto GO, BC and BC/GO-1.

Sample	Pseudo-First-Order Model			Pseudo-Second-Order Model			Experimental *q_m_* (mg/g)
	*K* _1_	*q_e_* (mg/g)	*R* _1_ ^2^	*K* _2_	*q_e_* (mg/g)	*R* _2_ ^2^	
BC	0.03982	208.8	0.9474	7.460 × 10^−5^	207.5	0.9757	121.0
BC/GO-1	0.04764	459.5	0.9505	3.529 × 10^−5^	398.4	0.9817	224.5

**Table 4 materials-17-03053-t004:** Adsorption isotherm parameters for Pb(II) adsorption onto BC and BC/GO-1.

Sample	Langmuir			Freundlich		
	*Q_max_* (mg/g)	*k_L_*	*R* _2_	*k_F_*	*n*	*R* _2_
BC	526.3	0.0029	0.6464	1.825	1.092	0.9983
BC/GO-1	1272.5	0.0024	0.6448	3.593	1.081	0.9944

## Data Availability

The original contributions presented in the study are included in the article, further inquiries can be directed to the corresponding author.

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
