# Peer review of "Pb(II) Adsorption Properties of a Three-Dimensional Porous Bacterial Cellulose/Graphene Oxide Composite Hydrogel Subjected to Ultrasonic Treatment"

_materials, 2024, doi:10.3390/ma17133053_

Round 1

Reviewer 1 Report

Comments and Suggestions for Authors

The manuscript describes a method for preparing composite hydrogels based on bacterial cellulose (BC) and graphene oxide (GO). Such systems are proposed to be used for lead removal. To describe the resulting composites, structural methods are used and the zeta potential is measured. The sources of lead and its harmful effects on the human body are noted. What makes the issues of removing lead from water, etc. relevant? The question immediately arises: how effective are systems based on activated carbon, porous carbon materials, etc.? After all, such systems are already used for water purification.

19. Small typo (paper, The adsorption...)
81. incorrect wording that needs to be corrected.
Section 3.1. It is advisable to start Characterization of BC/GO not with a drawing, but with text.
The signature for Scheme 1. - The process mechanism of BC/GO is not formatted correctly.
It is necessary to correct the caption for Fig. 1.
186. "maximum value" might be better replaced by "minimum value"
Table 2. It is necessary to check the name of the table.

In general, the manuscript, not counting typos and some methodological (terminology) inaccuracies, can be considered for publication. The conclusions of the manuscript reflect the findings.

Author Response

Dear Editor,

Thank you for your kind letter which about our article “Pb(â…¡) adsorption properties of three-dimensional porous bacterial cellulose/graphene oxide composite hydrogel subjected to the ultrasonic treatment(Manuscript ID: materials-3053789)” on June 05th,2024. We revised the manuscript in accordance with the reviewers' comments, and carefully proof-read the manuscript to minimize typographical, grammatical, and bibliographical errors.

Here below is our description on revision according to the reviewers' comments.

1.The reviewer's comment: How effective are systems based on activated carbon, porous carbon materials, etc.? After all, such systems are already used for water purification.

The authors' Answer: Compared with activated carbon and porous carbon, graphene oxide(GO) has a large specific surface area and abundant oxygen-containing functional groups on the surface, providing more adsorption sites for heavy metal adsorption.Meanwhile, it has been found that the composite of GO and activated carbon can further improve the adsorption capacity. Relevant references are listed below:

DOI: 10.1007/s11356-022-20343-6

DOI: 10.1039/d0en00193g

2.The review's comment: 19. Small typo (paper, The adsorption...)81. incorrect wording that needs to be corrected.

The authors' Answer: Corresponding grammatical errors have been corrected.

3.The reviewer's comment: Section 3.1. It is advisable to start Characterization of BC/GO not with a drawing, but with text.

The authors' Answer: The Section 3.1 is started Characterization of BC/GO with text.

4.The review's comment: The signature for Scheme 1. - The process mechanism of BC/GO is not formatted correctly.

The authors' Answer: The format of Scheme 1 have been corrected.

5.The review's comment: It is necessary to correct the caption for Fig. 1.

The authors' Answer: The problem has been corrected.

6.The review's comment: 186. "maximum value" might be better replaced by "minimum value".

The authors' Answer: The problem has been corrected.

7.The review's comment: Table 2. It is necessary to check the name of the table.

The authors' Answer: Comparison of maximum adsorption capacity of modified bacterial cellulose absorbents for Pb(â…¡).

Many grammatical or typographical errors have been revised.

All the lines and pages indicated above are in the revised manuscript.

Thank you and all the reviewers for the kind advice.

Sincerely yours,

Xinxing Zhang

Reviewer 2 Report

Comments and Suggestions for Authors

This work evaluated the adsorption properties of bacterial cellulose/ graphene oxide hydrogels. The work has some potential. However, some points listed below need to be improved.

Introduction: please add more related studies about Pb(â…¡) adsorption using bacterial cellulose and graphene oxide.

Introduction: clearer the novelty of this work.

Section 3.1: what characterization was done in this section? Scheme 1 is a “possible” process mechanism, because it is possible that “…BC will break the hydrogen bonds ( intramolecular hydrogen bonds and intermolecular hydrogen bonds ) and GO will be peeled off to single or less”.

Figure 2: I suggest use Absorbance instead of Adsorptivity. In addition, please presented the FTIR spectra from 4000 to 500 cm-1 and not from 500 to 4000 cm-1. Do the same in Figure 8.

Table 1: please better discuss the results presented in Table 1.

Lines 143-157: improve the discussion in this section and also add more references.

Section 3.4 and 3.5 have the same name. Please check.

I suggest also done desorption studies. It is possible reuse the hydrogel. How many utilization cycles were possible?

Author Response

Dear Editor,

Thank you for your kind letter which about our article “Pb(â…¡) adsorption properties of three-dimensional porous bacterial cellulose/graphene oxide composite hydrogel subjected to the ultrasonic treatment (Manuscript ID: materials-3053789)” on June 06th,2024. We revised the manuscript in accordance with the reviewers' comments, and carefully proof-read the manuscript to minimize typographical, grammatical, and bibliographical errors.

Here below is our description on revision according to the reviewers' comments.

1.The reviewer's comment: Introduction: please add more related studies about Pb(â…¡) adsorption using bacterial cellulose and graphene oxide.

The authors' Answer: We added a bit of research and references related to BC and GO adsorption.

2.The review's comment: Introduction: Clearer the novelty of this work.

The authors' Answer: We have added a statement of innovation in the last paragraph of the introduction.

3.The reviewer's comment: Section 3.1: what characterization was done in this section? Scheme 1 is a “possible” process mechanism, because it is possible that “…BC will break the hydrogen bonds ( intramolecular hydrogen bonds and intermolecular hydrogen bonds ) and GO will be peeled off to single or less.

The authors' Answer: Through ultrasonication, the hydroxyl group accessibility of BC was increased, which promoted the interaction with heavy metal ions; the introduction of GO, on the one hand, formed a looser three-dimensional structure with BC, which was conducive to the contact with heavy metal ions, and on the other hand, the C=O carried on the GO was easy to complex with heavy metal ions, which enhanced the ability of the bacterial cellulose to capture heavy metal ions.The mechanisms described in this section are obtained from the following analyses.

4.The review's comment: Figure 2: I suggest use Absorbance instead of Adsorptivity. In addition, please presented the FTIR spectra from 4000 to 500 cm-1 and not from 500 to 4000 cm-1. Do the same in Figure 8.

The authors' Answer: Figures 2 and 8 have been modified accordingly.

5.Table 1: please better discuss the results presented in Table 1.

The authors' Answer: We provide a more detailed description of Table 1.

6.The review's comment: Lines 143-157: improve the discussion in this section and also add more references.

The authors' Answer: The analytical discussion in the FT-IR chapter has been optimised.

7.The review's comment: Section 3.4 and 3.5 have the same name. Please check.

The authors' Answer: Section 3.5 is entitled Zeta potential studies.

8.The review's comment: I suggest also done desorption studies. It is possible reuse the hydrogel. How many utilization cycles were possible?.

The authors' Answer: Thank you very much for your suggestion, our follow up experiments are being planned for desorption studies as well as hydrogel life cycle studies.

Many grammatical or typographical errors have been revised.

All the lines and pages indicated above are in the revised manuscript.

Thank you and all the reviewers for the kind advice.

Sincerely yours,

Xinxing Zhang

Reviewer 3 Report

Comments and Suggestions for Authors

This paper describes the use of a BC/GO composite hydrogel, which demonstrated enhanced Pb(II) removal through increased availability of hydroxyl and carboxyl groups, achieving an adsorption capacity of 224.5 mg/g and showing potential as an alternative for heavy metal ion removal. However, before considering it for publication, the authors should address the following points:

Abstract

  • You have defined the acronym BC/GO twice; please correct this.
  • The nomenclature O(3)H is unclear: does it refer to the third hydroxyl group in the 3D structure?

Keywords

  • They are good, but I think you have room to add one more.

Introduction

  • How long does it take to synthesize bacterial cellulose? How can it be industrially viable to combine it with graphene oxide?

Materials and Methods

  • You should define GB/T, which is a national standard from China, as not everyone from other countries may recognize it.

Results and Discussion

  • In Figure 1c, the size mark is cut off; please fix this.
  • For Figure 2, how was the peak fitting of FT-IR done? Maybe I missed it, but please mention it.
  • "The carboxyl group stems from the unreacted group on GO, and the ester group is produced from the reaction of GO with BC." When they react, what by-product is generated?
  • On line 174, please reduce the number of decimal digits. Check this across the manuscript wherever it should be addressed.
  • In Figure 5, the font size is too small; please increase it. Check this across the manuscript wherever it should be addressed.
  • On line 216, "contact time" should be clarified.
  • In Table 2, the number of atoms in Fe3O4 should be in subscript. Check this across the manuscript wherever it should be addressed.
  • "The peak at 1733 cm-1 corresponding to C=O is not obvious, implying that this group interacts with lead ions." What do you mean by "not obvious"? It does not sound scientific; please explain this better.

Conclusion

  • Have you tested the composite for other heavy metals? If not, please provide a reason or plan for future studies.
  • Does this adsorbent work in different fields? Please elaborate further on possible contexts of application in the text.
  •  
Comments on the Quality of English Language

There are minor terminology issues that need to be addressed properly, as noted in my peer review. I invite the authors to thoroughly review the text again.

Author Response

Dear Editor,

Thank you for your kind letter which about our article “Pb(â…¡) adsorption properties of three-dimensional porous bacterial cellulose/graphene oxide composite hydrogel subjected to the ultrasonic treatment (Manuscript ID: materials-3053789)” on June 10th,2024. We revised the manuscript in accordance with the reviewers' comments, and carefully proof-read the manuscript to minimize typographical, grammatical, and bibliographical errors.

Here below is our description on revision according to the reviewers' comments.

1.The reviewer's comment: Abstract:·You have defined the acronym BC/GO twice; please correct this.·The nomenclature O(3)H is unclear: does it refer to the third hydroxyl group in the 3D structure?

The authors' Answer: One of the BGs/GOs has been deleted. O(3)H is the hydroxyl group in position three of the molecular structure.

2.The review's comment: Keywords:They are good, but I think you have room to add one more..

The authors' Answer: Thanks for the suggestion, we didn't find a more suitable sixth keyword at the moment.

3.The reviewer's comment: Introduction:How long does it take to synthesize bacterial cellulose? How can it be industrially viable to combine it with graphene oxide?

The authors' Answer: Depending on the processing time, it takes roughly 36-60 hours for BC to be processed into a BC dry gel.If we want to realise industrial production, we mainly need to solve the problems of industrial production and environmental protection of GO. In addition, it is necessary to strictly control the accuracy of the BC/GO preparation process.

4.The review's comment: ·You should define GB/T, which is a national standard from China, as not everyone from other countries may recognize it.

The authors' Answer: GB/T 1725-1979 is a Chinese executive standard on the method of determining the solid content of paints.

5.The review's comment: ·In Figure 1c, the size mark is cut off; please fix this..

The authors' Answer: The problem has been corrected.

6.The review's comment: ·For Figure 2, how was the peak fitting of FT-IR done? Maybe I missed it, but please mention it.

The authors' Answer: 

Table 1 Fitting results of hydrogen regional of FT-IR

Samples

Wavenumber/cm-1(Area IntgP %)

O(2)H···O(6)

O(3)H···O(5)

O(6)H···O(3)

BC

3566(13.7)

3436(46.42)

3228(39.88)

BC/GO-1

3563(9.55)

3432(44.01)

3251(46.44)

The FT-IR of Fig. 2 was fitted by hydrogen bonding of O(2)H---O(6), O(3)H---O(5), and O(6)H---O(3), and the results of the fit are shown in Table 1. to which we have added a note to Table 1.

7.The review's comment: "The carboxyl group stems from the unreacted group on GO, and the ester group is produced from the reaction of GO with BC." When they react, what by-product is generated?

The authors' Answer:In addition to shedding water molecules, dehydration reaction occurs between the hydroxyl group of BC and the phenolic hydroxyl group of GO, forming a C-O-C group.

8.The review's comment: On line 174, please reduce the number of decimal digits. Check this across the manuscript wherever it should be addressed.

The authors' Answer:The problem has been corrected.

9.The review's comment: In Figure 5, the font size is too small; please increase it. Check this across the manuscript wherever it should be addressed.

The authors' Answer:The problem has been corrected.

10.The review's comment: ·On line 216, "contact time" should be clarified..

The authors' Answer:"contact time" is the time of BC/GO and Pb(â…¡) adsorption.

11.The review's comment: ·In Table 2, the number of atoms in Fe3O4 should be in subscript. Check this across the manuscript wherever it should be addressed.

The authors' Answer:The problem has been corrected.

12.The review's comment: ·"The peak at 1733 cm-1 corresponding to C=O is not obvious, implying that this group interacts with lead ions." What do you mean by "not obvious"? It does not sound scientific; please explain this better.

The authors' Answer:The peak at 1733 cm-1 corresponding to C=O is much smaller than that without adsorption, implying that this group interacts with lead ions.

13.The review's comment: ·Have you tested the composite for other heavy metals? If not, please provide a reason or plan for future studies.

The authors' Answer:We are trying the adsorption effect of BC/GO for Cd2+, and would like to try Cr6+, Hg2+, As3+, etc. in the future.

14.The review's comment: Does this adsorbent work in different fields? Please elaborate further on possible contexts of application in the text.

The authors' Answer:The BC/GO materials in this study are more applicable to the treatment and remediation of heavy metals in soil or water quality. Some scholars have shown that BC/GO is a potential material for wound dressing due to its excellent biocompatibility and mechanical properties.

Many grammatical or typographical errors have been revised.

All the lines and pages indicated above are in the revised manuscript.

Thank you and all the reviewers for the kind advice.

Sincerely yours,

Xinxing Zhang

Round 2

Reviewer 2 Report

Comments and Suggestions for Authors

After corrections the manuscript reads well. I suggest publication in its current form.